



# Near-surface Palaeocene fluid flow, mineralisation and faulting at Flamborough Head, UK: new field observations and U-Pb calcite dating constraints

Nick M W Roberts[1], Jack K Lee[1, 2], Robert E Holdsworth[2], Christopher Jeans[3], Andrew R. Farrant[4],
Richard Haslam[4]

[1]Geochronology & Tracers Facility, British Geological Survey, Environmental Science Centre, Nottingham, NG12 5GG, UK
[2]Department of Earth Sciences, Durham University, Science Labs, Durham, UK
[3]Department of Earth Sciences, University of Cambridge, Downing Place, Cambridge, UK
[4]British Geological Survey, Environmental Science Centre, Nottingham, UK

*Correspondence to*: Nick M W Roberts (nirob@bgs.ac.uk)

**Abstract.** We present new field observations from Selwicks Bay, NE England, an exposure of the Flamborough Head Fault Zone (FHFZ). We combine these with U-Pb geochronology of syn- to post-tectonic calcite mineralisation to provide absolute constraints on the timing of deformation. The extensional Frontal Fault zone was active at ca. 63 Ma, with protracted fluid activity occurring as young as ca. 55 Ma. Other dated tensile fractures overlap this timeframe, and also cross-cut earlier formed fold structures, providing a lower bracket for the timing of folding and compressional deformation. The Frontal Fault zone acted as a conduit for voluminous fluid flow, linking deeper sedimentary units to the shallow sub-surface, and exhibiting a protracted history of several million years. Most structures at Selwicks Bay may have formed in a deformation history that is simpler than previously interpreted, with a protracted phase of extensional and strike-slip motion along the FHFZ. The timing of this deformation overlaps that of the nearby Cleveland Dyke intrusion and of regional uplift in NW Britain, opening the possibility that extensional deformation and hydrothermal mineralisation at Selwicks Bay are linked to these regional and far-field processes.

## 1 Introduction

Faulting of sedimentary basin fills in the subsurface is an important process in producing structurally constrained aquifers and reservoirs, as well as providing potential conduits and barriers to fluid resource migration and accumulation. Fault- and fracture-hosted infill and mineralisation allow us to assess the character and scale of along-fault fluid-migration. Maintenance of open fractures is an increasingly recognised phenomenon in faults formed in the shallowest parts of the crust down to depths of 1-2 km (e.g. Wright et al. 2009; van Gent et al. 2010; von Hagke et al. 2019). Open or partially open fractures can be propped open and preserved in the subsurface when they become infilled by wall rock collapse breccias, water-borne sediments and/or hydrothermal mineralisation (e.g. Walker et al. 2011; Holdsworth et al., 2019, 2020). These so-called fissure systems



have the potential to act as significant channelways for the migration and storage of subsurface fluids such as water, hydrocarbons or geothermal fluids, and in carbonate aquifers, can also act as pathways for the development of larger dissolutional condiuts and cave systems.

The absolute timings of fracture opening and fault displacement are critical to understanding how subsurface fluid migration evolves over time, and link individual fractures to the record of external tectonic deformation. Most sedimentary basins, whether ancient or currently active, lack direct chronological constraints on their structural history, and rely instead on the interpretation of stratigraphical and structural relations from field-data, or those imaged by geophysical means, e.g. seismic reflection data. Exposed faults can be directly dated if suitable geochronometers are preserved; recent methodological

developments have broadened the range of such mineral chronometers. Clay minerals can be dated by K-Ar, Ar-Ar and Rb-Sr, but require fault gouge, and a meticulous analytical approach to generate robust dates (e.g. Viola et al., 2016). U-Th/He dating of hematite coatings (e.g. Ault et al., 2016), U-Th-Pb dating of hydrothermal monazite (Bergemann et al., 2018) and Re-Os dating of hydrothermal sulphides (e.g. Dichiarante et al., 2016) are promising techniques that are also of use for faults and fault-hosted mineralisation of the right composition. In this paper, we utilise U-Pb dating of vein-filling calcite. Calcite is

an abundant material in brittle fractures and faults of wide-ranging host lithologies. It has been shown to be an effective chronometer that can be linked to the timing of hydrothermal mineralization, fault slip and fold development (Roberts & Walker, 2016; Ring and Gerdes, 2016; Goodfellow et al., 2017; Nuriel et al., 2017, 2019; Beaudoin et al., 2018; Hansman et al., 2018; Holdsworth et al., 2019, 2020; Parrish et al., 2018; Smeraglia et al., 2019; Roberts et al., 2020).

In ancient sedimentary basin systems worldwide, many episodes of uplift and deformation are a consequence of tectonic inversion associated with the far-field effects of orogenesis. In the British Isles, the youngest of these events is the Cenozoic (Neogene) Pyrenean-Alpine orogeny, which is linked to major geological structures exposed across Southern Britain (e.g. Chadwick, 1993; Blundell, 2002; Parrish et al., 2018), but may also have led to deformation as far north as Yorkshire, and offshore in the Southern North Sea (Ziegler, 1989). Here we present data from the Flamborough Head Fault Zone (FHFZ),

which forms the southern boundary to the Mesozoic Cleveland Basin, and to which there is no consensus as to the timing and kinematic history. In this paper, we combine new field observations with U-Pb geochronology of calcite veins. Our dates present the first absolute timing constraints on deformation within the FHFZ, and are placed into context with new field observations pertinent to understanding the structural setting of associated fluid flow and fracture filling processes.

## 60    2 Geological Setting

The Mesozoic Cleveland Basin (Fig. 1a) located in East Yorkshire, northern England, has experienced inversion the timing of which is poorly constrained. It is generally attributed by most authors to distant effects of the Pyrenean-Alpine orogeny (e.g. Starmer, 1995). The Jurassic-Cretaceous basin fills are bounded to the north and south by complex fault zones. To the south, the FHFZ is an east-east striking zone of brittle faults, which separates the Cleveland Basin from the Market Weighton Block



to the south (Fig. 1a). Inland exposures of the fault zone are poor, and largely restricted to small quarries in Cretaceous chalk; however, they can be mapped on the surface, and are visible on remotely sensed datasets (Farrant et al., 2015). In contrast, the coastline preserves excellent exposures of the faults and associated deformation. Flamborough Head (Fig. 1b) exposes several fault zones that have a complex structure and potentially a protracted history; these are the Bempton, Selwicks Bay, and Dykes End fault zones (Fig. 1b).


The FHFZ is an E-W zone of brittle faults exposed at the coast at Flamborough Head, and extending inland for 30 to 40 miles (see Fig. 1a, and Farrant et al., 2015). The fault zone is linked with the Vale of Pickering Fault Zone (Kirby and Swallow, 1987), and has also been referred to as the Howardian Hill-Flamborough Fault Belt (Starmer, 1995). To the east, the fault zone is truncated offshore by the Dowsing Fault Zone, which forms the western margin of the Sole Pit Basin. The deformation of

the Cretaceous chalk rocks around Flamborough Head associated with some of the E-W faults has long been studied due to the excellent and structurally complex exposures preserved here (e.g. Phillips, 1829; Lamplugh, 1895; Kent, 1974; 1980; Kirby & Swallow, 1987; Peacock & Sanderson, 1994; Starmer, 1995, 2008, 2013; Rawson & Wright, 2000; Sagi et al., 2016).

Previous geological constraints on the timing of fault movements within the FHFZ come from the interpretation of seismic

reflection data and sedimentological and structural analyses of several key outcrops (Jeans, 1973; Kirby and Swallow, 1987; Starmer, 1995). The offshore seismic interpretation of Kirby and Swallow (1987) indicates the existence of both steep faults that cut underlying Permian and Carboniferous strata at depth, and listric faults that detach within the Permian Zechstein strata. Faults on the northern and southern margins of the fault zone form a graben structure. Thickness changes of the Speeton Clay and Red Chalk have been interpreted as evidence that the northern fault zone (comprising the Bempton and Speeton faults;

Fig. 1b) began in the early Cretaceous (Jeans, 1973; Neale, 1974; Kirby and Swallow, 1987). Kirby and Swallow (1987) concluded that an early stage of near-vertical normal faulting (early Cretaceous) produced the graben structure, which was then followed by a period of listric normal faulting, with both events occurring prior to the Late Cretaceous. Inversion of the former extensional structures, forming the 'Shatter Zones' at Bempton and Selwicks Bay, is inferred to have occurred at the end of the Cretaceous, and has been related to the more regional uplift, folding and inversion of the Cleveland Basin to the

north forming amongst other structures, the Cleveland anticline (Fig. 1a, Kirby et al., 1987).

Peacock and Sanderson (1994) conducted a detailed investigation of the orientation and displacements of faults exposed around Flamborough Head, covering some 1340 individual structures. They interpret their data as indicating that sigma1 during faulting was sub-vertical, with extension occurring sub-horizontally in all directions, and that complex relationships existed

between sigma2 and sigma3. Based on oblique-slip kinematics they suggested that a sub-horizontal sigma3 developed over time in a dominantly NNW-SSE direction. These authors also briefly describe the existence of contractional structures, namely oblique or reverse displacements on some fault surfaces, with a NNW-SSE contraction direction. Brecciation and veining are





pervasive at Selwicks Bay, and Peacock and Sanderson (1994) tentatively suggest that both were related to the contractional event. They do not, however, present clear evidence for whether contraction preceded or followed extension.


Starmer (1995) produced a deformation history of the chalk at Selwicks Bay based on detailed mapping and structural analyses in onshore exposures of chalk; he describes four phases of deformation (D1 to D4). D1 produced folds with NNW-SSE axes and bedding plane-parallel shears, with an ENE-WSW to E-W compression direction. Subsequent D2 deformation was attributed to extension in an E-W direction, and the formation of tensional extensional fractures. D3 started with E-W directed

flexure of the strata, and some strike-slip faulting. Once the folding had tightened, thrusts with top-to-the-S to -SSE shear sense directions cut through the strata. Dextral shearing at the same time suggested an element of transpression alongside the dominant N-S compression. D4 was interpreted as a complex phase of extension-transtension, first with E-W extension allowing N-S structures to activate, and followed by a N-S component of extension. Starmer (1995) links these events to Laramide (late Maastrichtian to early Palaeocene) compression (D1), Laramide extension (D2), Alpine (Oligocene)

compression (D3) and post-Alpine extension (D4).

Sagi et al. (2016) studied the exposures at both Selwicks Bay and Dykes End (see Fig. 1b), analysing the fault density and connectivity in particular, and the relationship of these to fluid-flow. These authors describe numerous occurrences of dilational and contractional jogs occurring along the fault planes, exhibiting textures that include pressure solution styolites and coarse-

crystalline calcite vein-fill. At Selwicks Bay, Sagi et al. (2016) focused on two large, steeply-dipping ENE-WSE normal faults that are a distinctive structural feature (the Frontal Faults of Starmer, 1995), related to folding and intense metre scale zones of intense veining and brecciation in the chalk wall rocks. These authors describe the damage zones associated with this set of faults (the Selwicks Bay 'Shatter Zone') and show that they are 4 to 5 m wide in the footwall, but less than 1 m wide in the hanging-wall. These brecciated damage zones are where the highest intensity of veining occurs forming a highly

interconnected, braided network of tensile calcite-filled fractures. This study showed that fluid connectivity was much higher in the damage zones of the faults (up to 60%) compared the surrounding protolith (less than 10%), i.e. that these faults unequivocally represented highly effective fluid conduits in the geological past.

Faÿ-Gomard et al. (2018) present a geochemical study of veining at Selwicks Bay, and describe a relative chronology of three

different phases of calcite veining. They use clumped isotopes of carbonates to determine precipitation temperatures of ca. 60°C, and combined with carbon, oxygen and strontium isotope analyses, postulate that the fluids originate from the underlying Triassic Sherwood sandstone. These authors link the timing of veining to Late Cenozoic to Cenozoic basin inversion, suggesting veining may have occurred in a pulsed manner in occurrence with pulsed phases of inversion; although in their figure, they utilise the burial history curve of Emery (2016), correlating the veining with Oligocene-Miocene regional uplift.






Mortimore (2019) revisited the stratigraphy of the chalk exposed at Selwicks Bay, providing new stratigraphic and sedimentological logs for exposures north and south of the Frontal Faults. Mortimore (2019) also re-evaluated both micro and macro–scale structures and sedimentological features exposed at Selwicks Bay, questioning whether many of those exposed are a result of syn-sedimentary slumping and downslope displacement, rather than being purely tectonic processes.

## 3 Methodology

Fieldwork focussed on examples of calcite mineralization associated with folds, fractures and faults in the well-exposed Selwicks Bay (Fig. 1c). The chalks here are amongst the youngest exposed in Yorkshire and include the older Burnham Chalk Formation (Upper Turonian to Coniacian) overlain by the younger Flamborough Chalk Formation (Santonian to Campanian) (see Whitham, 1993). Several samples of calcite mineralization were collected for U-Pb dating purposes, with additional material collected from some fracture fills in order to understand the geological context of fracture-filling processes. Thin sections from these samples were studied optically in order to characterise the mineralogy, structural setting and – where possible – the sequence of fracture filling in each sample.

U-Pb geochronology was conducted at the Geochronology and Tracers Facility, British Geological Survey, UK. Samples were analysed using polished epoxy blocks/slabs. The instrumentation used was a New Wave Research 193UC excimer laser ablation system fitted with a TV2 cell, coupled to a Nu Instruments Attom single collector inductively coupled plasma mass spectrometer (ICP-MS). The method follows that described in Roberts et al. (2017). Laser parameters were pre-ablation conditions of 150 µm static spots fired at 10 Hz with a fluence of ~8 J/cm$^2$ for 2 seconds, and ablation conditions of a 100 µm spot, fired at 10 Hz with a fluence of ~8 J/cm$^2$ for 30 seconds. A 60 second background is taken before every set of standard-bracketed analyses, and a 5 second washout is left between each ablation. Data reduction uses the Time Resolved Analysis function of the Nu Instruments Attolab software, and an excel spreadsheet. Isoplot v4 (Ludwig, 2011) is used for calculation and plotting of ages. Uncertainty propagation follows the recommendations of Horstwood et al. (2016) and all dates include propagation of systematic uncertainties. The carbonate material WC-1 (254 Ma; Roberts et al., 2017) was used as the primary reference material, and Duff Brown Tank (64.04 Ma; Hill et al., 2016) and ASH15D (2.96 Ma; Perach Nuriel pers. Comm. 2020) were used as validation materials. The pooled result of all Duff Brown analyses yields a lower intercept age of 65.4 ± 1.2 Ma, and the pooled result of ASH15D yields a lower intercept age of 2.88 ± 0.08 Ma.

## 4 Field observations and outcrop settings of samples

It is not our intention here to provide a detailed kinematic analysis of faulting in the region, instead, it is our objective to simply provide the context for our new U-Pb dates in terms of the general movement history of the fault zones and the associated hydrothermal mineralisation.





Three samples (NR1707, NR1708 and CJ1) come from the brecciated regions associated with a sub-parallel set of steeply northward dipping normal faults exposed at the south of Selwicks Bay (Fig. 2a; the E-W 'Frontal Faults' of Starmer (1995), and the Intensely Brecciated Zone (IBZ) of Sagi et al. (2016). The combined displacement across these faults has been estimated as ~20 m based on stratigraphic offsets (Rawson and Wright, 2000; Mortimore, 2019), with downthrow to the north. Either side of the fault zone – and particularly in the hangingwall region – cm- to m-scale drag folding of the beds in the chalks clearly demonstrates this sense of motion (Fig. 2a). The two faults separate a 4-5 m wide zone of highly calcite veined, variably misoriented and brecciated chalk (Fig. 2a-2d). The areas of breccia are highly variable in their development – some smaller examples up to 20 cm wide are fairly constant in thickness and are bounded by well-defined planar fracture surfaces (Fig. 2d), whilst others are more irregular, with diffuse margins, varying between a few cm to more than 1.5 m across. As noted by Sagi et al. (2016), main veins in the brecciated panel between the two bounding faults show geometries consistent with tensile (mode I) opening during normal faulting (Fig. 2d) with the development of well-defined median lines and, in places, open vuggy cavities suggesting syntaxial mineralization into large open voids (Woodcock et al., 2014).

Another sample, NR1709 comes from an E-W steeply dipping tensile calcite vein in the heavily fractured natural pavement close to the base of North Cliff (Fig. 1c). The structural setting of these veins is seen in the cliffs immediately along strike and to the west where a steeply S dipping normal fault with dip-slip slickenlines is seen offsetting an earlier low angle thrust fault with a cm-scale, close to tight southward verging antiform in its immediate hangingwall (Fig. 3a). Close to the beach at the base of the cliff, the E-W subvertical calcite veins are seen to be well developed in the immediate hangingwall of the normal fault and show a sense of obliquity consistent with the normal shear sense along the fault (Fig. 3b), suggesting that they are the same age.

A final sample, NR1901 comes from a tight synformal fold some 40m N of the northern Frontal Fault (Fig. 1c). Here a metre-scale, close to tight, southward verging antiform-synform pair is developed and is closely associated with at least two top-to-the-S low to moderately N dipping thrust faults (Fig. 3c). The exposed synformal fold hinge reveals bedding-parallel calcite slickenfibres oriented at high angles, oblique to the fold hinge (Fig. 3d) consistent with the operation of oblique flexural slip processes during folding (e.g. Holdsworth et al., 2002). The age of the folds relative to the normal movements along the Frontal Faults is unclear as no clear mesoscale cross-cutting relationships are seen, but the folds and thrusts cross cut by the normal faults at North Cliff are identical in style to those at this last locality. Thus, it is suggested that the main phase of extensional displacement and hydrothermal calcite mineralisation associated with the Frontal Fault Zone likely post-dates an earlier phase of generally top-to-the-S thrusting and folding.

## 5 Fracture fills and microstructure

The contractional and extensional phases of deformation seen in Selwicks Bay are associated with significantly different fault rocks and fracture fills.






## 5.1 Contractional structures

The earlier low angle thrusts and folds are typically marked by narrow (<5cm thick) zones of incohesive crush breccia and gouge (Fig. 4a and 4b), with gouges often best developed where thrust faults interact with clay-rich 'marly' interbeds in the chalk. Local gouge injections <1mm thick are seen cutting the wall rocks adjacent to thrusts (Fig. 4b). Calcite mineralization
is largely limited to the development of slickenfibres along exposed thrust planes (Fig. 4c) and bedding planes around metre-scale folds (Fig. 4d). These slickenfibres show widespread evidence for crack-seal textures and are locally cross cut by later veinlets of structureless sparry calcite (Fig. 4e-4f).

## 5.2 Extensional structures

The fracture fills associated with both the 'Frontal Fault zone of Starmer (1995) and the small scale normal faults and associated veins elsewhere in Selwicks Bay are significantly different compared to the earlier contractional features.

Shear fractures have various orientations in the wall rocks, comprising small-offset (< 0.5m) normal faults with dip-slip slickenlines (Sagi et al., 2016). These are closely associated with steeply dipping to subvertical generally E-W trending calcite
veins filling tensile (Mode I) fractures (e.g. Figs 2b, 3d, 5a-c; the Group I veins of Faÿ-Gomord et al. 2018). Fills are predominantly fine to coarse-grained sparry calcite and commonly form as braided, up to 0.5 m wide zones of veins (e.g. Fig 2b) that resemble the "zebra rocks" described by Holland and Urai (2010) in low porosity limestones in Oman. Most individual veins have an average thickness of 1–2 mm, but the thickest can (locally) reach widths of up to 30 cm. Many veins are composite having more than one calcite fill with subtle differences in colour.


Breccia fills are mostly associated with the Frontal Fault zone. The majority are generally E-W to ENE-WSW trending, steeply dipping, with clasts dominated by chalk that are clearly derived from the host wall rocks, although differences in texture and colour of individual clasts relative to immediately adjacent wall rocks and other clasts indicate a degree of mixing and displacement from source. The breccias show every gradation from incipient crackle (Fig. 5a) through mosaic to chaotic
textures (Figs 5b, c), with clasts becoming generally more rounded as the fill becomes chaotic (Woodcock & Mort, 2008). Importantly, the fills show very little evidence for shearing or attrition of clasts and closely resemble collapse breccias formed by wall rock collapse and infilling into open tensile fissures in near surface faulting environments (Woodcock et al., 2006; Holdsworth et al., 2019).

The breccia matrices are compositionally very variable. Some are clay rich ('marly') and darker coloured whilst the majority are lighter coloured with less clay and are well cemented by sparry calcite (the Group II veins of Faÿ-Gomord et al., 2018). Generally, E-W trending calcite veins essentially identical to those seen in the wall rocks are seen to both cross-cut breccia as





well as being included as clasts in breccia or as earlier misoriented veins cross-cutting chalk clasts (Fig. 5b). This might suggest that calcite mineralization, breccia formation and cementation were broadly contemporaneous processes. Sample NR1707 in
the current study is taken from a matrix cement, while NR1708 is from an earlier vein that cuts a chalk clast. Many veins are composite having more than one calcite fill with subtle differences in colour; weathering on the foreshore reveals both ferroan calcite (stained red due to oxidation) and non-ferroan calcite (unstained) fills (Fig 5c, implying changing fluid chemistry during the period of fracture-fill mineralisation.

A notable features of the locally later tensile calcite vein fills (Group III veins of Faÿ-Gomord et al., 2018) is the widespread development of vuggy cavities (Figs. 2d and 5b); these are particularly widespread in the Frontal Fault zone. Their development implies that in the latter stages of vein filling at least, rates of mineral precipitation were reduced relative to fracture opening rates, implying that fractures remained open for protracted periods of time. Sample CJ1 comes from one of these large vuggy vein fills.


Further evidence for the development of long-lived open fissures in the Frontal Fault zone comes from the preservation of brown-coloured marly breccias and sediment fills in tensile fissures (Figs. 6a-d). These occur as sub-vertical features that both post-date and predate adjacent sub-parallel calcite veins (Figs. 6a and b, respectively) and in steeply inclined fissures that obliquely cross-cut adjacent veins (Fig. 6c). More rarely, irregular subhorizontal zones of fine marly sediment fill the lower
part of fractures that cross-cut earlier calcite veins, whilst the upper part of the cavity is filled with later calcite (Fig. 6d). These sediments are crudely bedded and represent geopetal structures that consistently young upwards wherever they are found.

Thin sections reveal that the majority of calcite veins are syntaxial and sparry (Fig. 7a). The marly breccias and sediment fills contain numerous fragments of wall rock chalk, earlier calcite vein fills and more exotic materials such as brown clays, chert,
individual microfossils - including sponge spicules - and rounded grains of both quartz and glauconite (Fig. 7b-d). The geopetal fills preserve striking examples of graded bedding (Figs. 7a and 7e) and cockade style mineralization textures (Fig. 7f), with fine grained, graded suspensions of sedimentary grains floating in single crystals of calcite cement grown in perfect optical continuity with adjacent vein fills (Figs. 7a, 7e, and 7f). The preservation of such features suggest that sedimentary material was transported by flowing fluids into open cavities connected to the surface and that cementation associated with
contemporaneous hydrothermal mineralization froze the finer materials in place before they were able to settle out of suspension (cf. Wright et al., 2009; Frenzel & Woodcock, 2014).

In summary, most, but not all of the calcite mineralization seen at Selwicks Bay is related to extensional structures that locally appear to post-date an earlier phase of cm to m-scale top-to-the-S folding and thrusting. Mineral veins are predominantly
tensile and generally E-W trending and appear to be broadly contemporaneous with the development of calcite mineralized breccias along the Frontal Fault zone. The breccias preserve widespread textures consistent with wall rock collapse into open





cavities rather than being the product of attritional cataclasis. The existence of long-lived open fissures is confirmed by the widespread preservation of vuggy textures and cockade-style calcite mineralization, together with the local development of marly sediment fills and geopetal structures. Based on the large amount of calcite mineralization – especially along the Frontal

Fault zone, it is clear that substantial volumes of fluid flow have been localised along this fault zone during extension (Sagi et al. 2016).

## 6 Results

Tera-Wasserburg plots of the resulting U-Pb data are shown in Figure 8. NR1707 yielded a lower intercept date of 63.9 ± 2.6 Ma (MSWD = 2.1); this date is from seventeen spots from one crystal. NR1708 yielded a lower intercept date of 63.4 ± 5.3

Ma (MSWD = 1.8); this date results from a traverse of one crystal comprising forty-nine spots. CJ1 is from a localised region of one large calcite crystal, towards its base; sixty-one spots yielded a date of 54.9 ± 3.1 Ma (MSWD = 1.5). NR1709 yielded a lower intercept date of 56.2 ± 8.2 Ma (MSWD = 1.6 Ma); this result was obtained from two crystals, comprising fifty-one spots in total. Two domains of NR1901 were calculated separately. The first domain comprising thin (<200 µm) layers of slickenfibre calcite yielded no reasonable date, as the data are dominated by common lead (see supplementary file). The second

domain comprising a cross-cutting veinlet yielded a date of 58.8 ± 1.9 Ma (MSWD = 1.4); this date is from seventy-three spots. The five successful dates provide a spread in crystallisation of nine million years, although taking uncertainties into account, this may be as small as three million years. The three samples from the Frontal Fault (NR1708, NR1709, CJ-1) do not overlap when considering their age uncertainties, indicating a protracted period of fluid-flow of several Myrs.

## 7 Discussion

### 7.1 The timing of deformation at Selwicks Bay

The dates obtained from the five samples yield constraints on the timing of deformation at Selwicks Bay. NR1707 and NR1708 are inferred to directly date the extensional phase of deformation along the Frontal Fault zone (and FHZF) as they are from regions of calcite veining and cemented collapse breccias. These samples provide overlapping ages of 63.9 and 63.4 Ma. CJ1 is also from the Frontal Front zone, but yields a younger age of 54.9 Ma that is outside of analytical uncertainty of the breccia

samples. This younger date is from a relatively late large vuggy fracture fill. We cannot be certain whether this younger date reflects a regionally later fracture opening event, but there are no clear field or thin section relationships observed to suggest this. NR1709 has a large uncertainty, overlapping both the breccia and younger vuggy calcite from the Frontal Fault. The date indicates that fracture opening at the northern part of Selwicks Bay overlaps that of the Frontal Fault in the southern part of the bay. The dated late veinlet within sample NR1901 overlaps the dates of the other samples (except NR1707). Since this

veinlet cross-cuts the slickenfibre growth, this date provides a lower boundary for the timing of the folding and associated cocontractional/transpressional deformation.



## 7.2 Implications for chalk-hosted fluid-flow

Chalk is an important aquifer for groundwater, particularly in parts of Britain and surrounding countries in Europe (e.g. Price, 1987; MacDonald and Allen, 2001). Chalk can also act as both reservoirs and seals for hydrocarbons (e.g. Hardman, 1982;
Mallon & Swarbrick, 2008). As such, the timing and origin of fracture-hosted permeability is an important constraint on understanding fluid-flow through chalk.

The Frontal Fault zone structure at Selwicks Bay represents a significant damage zone associated with normal faulting in the region. This fault zone forms part of the FHFZ, but has much less offset than other fault-zones to the north (Bempton Fault)
and south (Langtoft Fault) (Fig. 1a-1b). It is clear, however, that the large fissure systems forming the fault zone have acted as a major fluid conduit allowing voluminous fluid-flow through the chalk, possibly over a long time period of at least five million years. Interestingly, geochemical analyses of the calcite fills by Faÿ-Gomord et al. (2018) show that all the calcite veins share the same chemical signature which they link to an underlying source of meteoric fluids in the Triassic Sherwood Sandstone. Salinities vary, suggesting some mixing with saline fluids. Given the development of open vugs and geopetal sediment fills
with glauconite and microfossil fragments, a link to a surface marine environment is indicated at the time of calcite mineralization. The development of contemporaneous open fissures with sediment infilling due to wall rock collapse and washing in of finer materials from the surface, together with hydrothermal mineralization from below during tectonic extension is an increasingly recognised phenomenon in near surface fracture systems (< 1-2km depth; e.g. Wright et al. 2009; Walker et al., 2011; Holdsworth et al. 2019, 2020).

The fault has acted as a 'fluid superhighway' connecting deeper reservoir units (Triassic sandstones) with the surface during the latest Cretaceous-earliest Palaeocene. The existence of a fluid conduit of this kind potentially has major implications for storage and migration processes associated with reservoirs, whether they be for groundwater or hydrocarbons. Importantly, this structure is potentially of sub-seismic scale, indicating that even sub-seismic features may host large-scale fluid-flow, and
produce significant conduits that exhibit high permeability over protracted time periods lasting millions of years. We also point out that this is just one small fault of many in the FHFZ, and that many of the faults exposed inland are also associated with extensive veining, as well as secondary cementation of the chalk adjacent to the faults. These secondary cements form hard chalk zones, which then potentially act as barriers to fluid-flow.

## 7.3 Implications for regional tectonics

The Flamborough Head Fault Zone forms a structural boundary that separates the Cleveland Basin to the north, and the Market Weighton Block to the south (Kirby & Swallow, 1987; Starmer, 1995). The history of the fault zone is thought to be influenced by the subsidence and later inversion of the Cleveland Basin, whilst the Market Weighton Block remained high and stable (Kent, 1980). The Flamborough Head Fault Zone is truncated to the east by several intersecting deformation zones (Central Fracture Zone, Dowsing Fault Zone, Sole Pit Basin; see Fig. 1), and truncated onland by the Humanby Trough-Peak Fault





zone (see Fig. 1 and Ford et al., 2020). The deformation that led to the formation and inversion of these basins has a long
history extending from the Permo-Triassic to the Miocene (e.g. Starmer, 1995 and references therein), and the far-field stress
associated with their formation may have some relevance to the Flamborough Head Fault Zone. The histories of these offshore
regions are only constrained by seismic and borehole data, and correlation with known regional events. Therefore, dating of
onshore structures such as those presented in the current study provides additional and new absolute timing constraints on the
structural evolution at a regional scale.

There have been long-standing differences in the interpretations of the structural complexity of deformation in the
Flamborough Head region. Some prefer a polyphase deformation sequence over a protracted time period from the later
Cretaceous to Neogene times (e.g. Starmer, 1995 and references therein) whilst others favour a somewhat simpler regime
involving shorter-lived periods of strike-slip tectonics and polymodal, possibly polygonal extensional faulting (e.g. Peacock
& Sanderson, 1994; Sagi et al., 2016, Faÿ-Gomord et al., 2017).

Our findings show that a regionally significant extensional phase of deformation occurred over a protracted period during very
latest Cretaceous to early Eocene times (ca. 64-55 Ma). We suggest that this represents the youngest phase of deformation
seen along the Frontal Fault zone – and by inference the FHFZ - post-dating any contractional or transpressional deformation.
It should be noted that we cannot rule out that fluid-flow and tensile fracturing may have extended to even younger dates than
our study implies. Interestingly, this timing of deformation overlaps but is broadly younger than the estimated late Cretaceous
timing of widespread inversion and tectonic events across parts of NW Europe, discussed by Mortimore (2018).

The age ranges for calcite mineralization overlap almost exactly with the timing of igneous activity in W Scotland and Northern
Ireland related to mantle upwelling activity forming the British Paleogene Igneous Province (Jolley & Bel, 2002), and
associated regional uplift (Lewis, 2002; Nadin et al., 1997). In this regard, a clear geological link exists given the presence of
the nearby Cleveland Dyke, the easternmost exposure of which lies some 30 km NW of Selwicks Bay (Fig 1a). The intrusion
of this dyke – which can be traced across a wide region of northern Britain, is thought to be ca. 58-55 Ma based on K-Ar dating
(Fitch et al., 1978; Evans et al., 1973). Our findings open up the possibility that that extension and associated fluid flow in the
Flamborough Head region are related to the far field influence of N Atlantic opening processes.

Our findings further suggest that folding and thrusting of the chalk at Flamborough Head must be older than ca. 64 Ma. Given
the Santonian to Campanian age of the youngest chalk affected by deformation (ca. 86-72 Ma; Whitham, 1993; Mortimore,
2019), this implies that the inversion event can be no older than latest Cretaceous. In previous interpretations, much of the late
stage compressional deformation along the FFHZ has been linked to inversion related to the far-field effects of the Alpine
orogeny during the Neogene (Starmer, 1995). Clearly, our findings from Selwicks Bay cast significant doubt on this model. It
seems possible that the earlier folding and thrusting seen at Selwicks Bay and elsewhere around Flamborough Head is related



to a phase of strike-slip deformation along the FHFZ. Based on our findings to date, we cannot rule out the possibility that
these strike-slip events overlap with the later extensional deformation, i.e. they are all manifestations of a protracted phase of
regional transtensional tectonics in latest Cretaceous to Palaeocene times. Thickness changes in the chalk around the faults
exposed at Flamborough Head (see Mortimore, 2019 and references therein), are the only evidence for extensional deformation
earlier than our oldest date of 64 Ma.

We propose that a reassessment of the deformation structures and sequences in the onshore and offshore regions around
Flamborough head is required, ideally with further absolute dating and palaeostress inversion analyses. More generally, our
findings are a further illustration that the sequence, timing and tectonic significance of the Cenozoic history of the British Isles
may be in need of some reassessment (e.g. see discussion in Parrish et al., 2018).

## 8 Conclusions

U-Pb dating of calcite vein-fill from Selwicks Bay provides constraints on the timing of faulting. Five dates, ranging from 63.9
to 54.9 Ma, indicate that formation of the mineralized collapse breccia within the extensional Frontal Fault zone occurred at
ca. 63 Ma, with fluid-flow continuing to at least 55 Ma. Calcite from a Mode I tensile vein in the nearby wall rocks has a large
age uncertainty but overlaps both these dates. A veinlet cross-cutting slickenfibres formed on a bedding parallel surface of a
fold structure, places a lower boundary on folding at 56 Ma. The dates indicate that faulting within the Flamborough Head
Fault Zone was Palaeocene in age. We dispute a compressional (and tectonic inversion) origin for most structures at Selwicks
Bay, instead suggesting that, except for the possibility of syn-sedimentary slump structures, a more straightforward model
involving overlapping strike slip and extensional deformation may explain all of the deformation. Our study has shown that
the extensional Frontal Fault zone at Selwicks Bay represents: [1] a fault-hosted fluid conduit that linked deeper sedimentary
units to the shallow sub-surface, and hosted voluminous fluid-flow over a protracted time-scale; and [2] its fault activity
occurred within a 5-10 Ma time frame overlapping with that of the intrusion of the nearby Cleveland Dyke (ca. 58-55 Ma), the
development of the N Atlantic Igneous Province and the regional uplift of NW Britain related to the opening of the North
Atlantic.

## Acknowledgements

NR, AF and RH publish with the permission of the Executive Director of the British Geological Survey. The authors thank
Mark Woods for discussion, and constructive reviews from xx.



**Code and data availability.**

U–Pb data presented in the Supplement Table, along with the corresponding methods and analytical details in the Supplement Text.


**Supplement.**

The supplement related to this article is available online at:

**Author contributions.**

NMWR and JKL collected the analytical data. NMWR, AF, JKL, CJ and REH conducted fieldwork and sample collection. NMWR, JKL and REH conducted sample imaging and petrography. All authors contributed to writing the paper.

**Competing interests.**

The authors declare that they have no conflict of interest.


**Financial support.**

This research has been supported by the UK Natural Environment Research Council (grant no. NE/S011587/1).

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





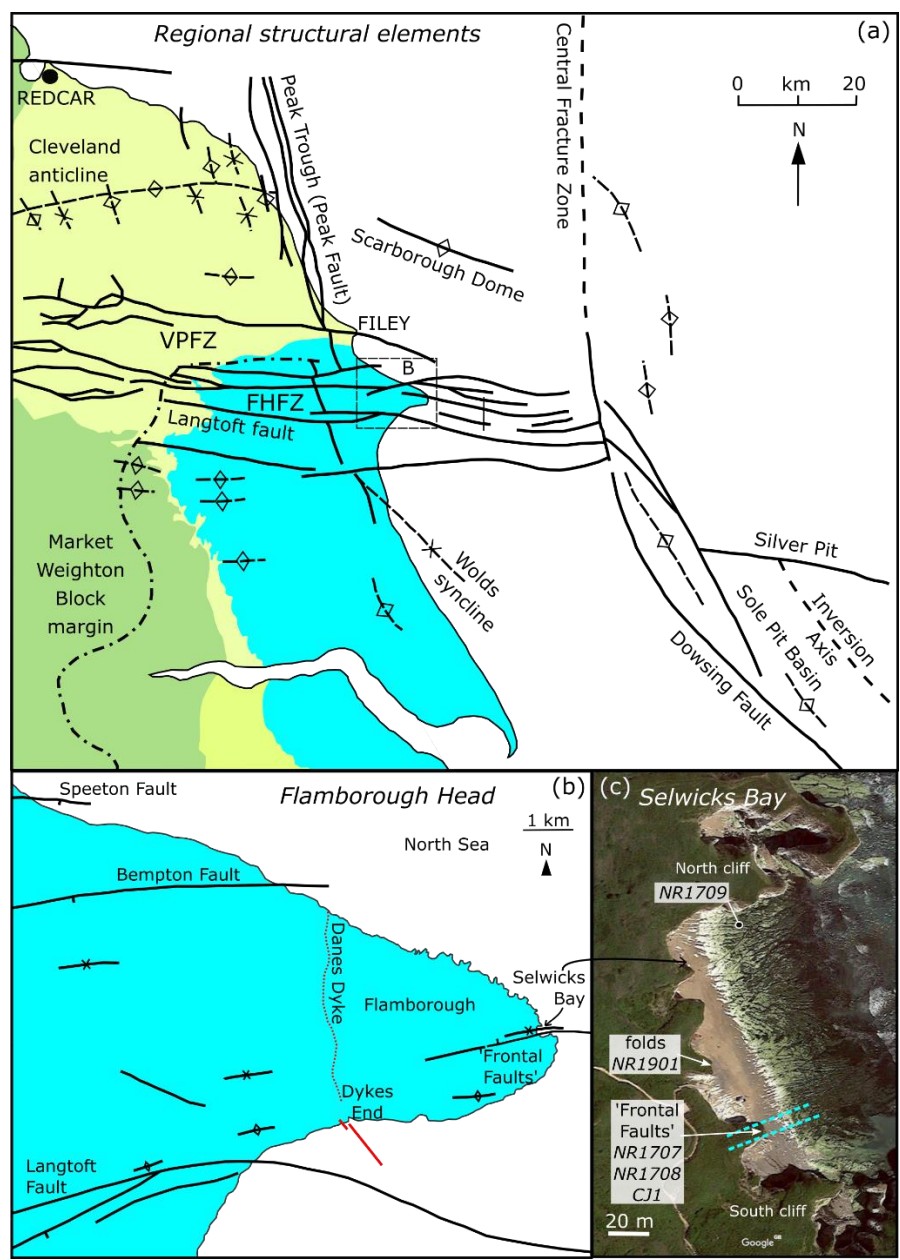

Figure 1. (a) Geological sketch map of the region around Flamborough Head, NE England, showing the regional structural elements. Modified after Powell (2010) and Starmer (1995), reproduced by permission of the Council of the Yorkshire

550 Geological Society. (b) Geological sketch map of Flamborough Head showing main structural features. Modified after Starmer (2013), reproduced by permission of the Council of the Yorkshire Geological Society. (c) Satellite image of Selwicks Bay showing location of samples. Google map data: Imagery ©2020 CNES / Airbus, Getmapping plc, Infoterra Ltd & Bluesky, Maxar Technologies, Map data ©2020.

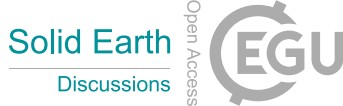

555

Figure 2. (a) The Frontal Faults North and South (red dashed lines labelled FFN and FFS, respectively) viewed looking west in the cliffs at Selwicks Bay. The blue dashed line indicates bedding in the chalk either side of the fault zone, showing a prominent zone of drag folding consistent with a north-side down sense of relative motion along the fault zone. The locations of the images shown in b-d are also shown. (b) Relatively planar fault zone with breccia that forms part of the FFS. Note gentle drag folding in both hangingwall and footwall consistent with N-side down motion. (c) Wider, more irregular fault breccia that forms part of the FFN, with clasts of wall rocks up to 1.5m across. (d) Open vuggy tensile vein with partial sparry calcite fill from brecciated region bounded by FFS and FFN. Note the opposite dip to the bounding faults consistent with N-side down motion.

Figure 3. (a) View looking W of top-to-the-S thrust fault (red) cross-cut by S-side-down, steeply dipping normal fault (black), north side of Selwicks Bay. Bedding in the hangingwall (blue) of the thrust is deformed by a cm-scale, S-overturning anticline. Box shows location of (b). (b) Close up view of normal fault shown in (a) with N-dipping tensile veins filled with calcite in hangingwall consistent with S-side-down sense of throw. (c) View looking W of metre-scale, close to tight, southward verging antiform-synform pair and associated top-to-the-S low to moderately N dipping thrust faults some 40m north of the FFN. (d) View looking S of exposed synformal fold hinge shown in (c) with bedding-parallel calcite slickenfibres oriented oblique to the fold hinge.





Figure 4. Structures associated with contractional structures in Selwicks Bay. (a) Crush breccia along top-to-the-S thrust fault reoriented by later, normal fault-related tilting. (b) Crush breccia and brown gouge derived from chalk and shale, respectively, associated with top-to-the-S thrust fault with narrow gouge injections into footwall, one of which is arrowed.

(c) Calcite-hematite slickenlines associated with top-to-the-S thrust fault. (d) Close-up of stepped, bedding parallel calcite slickenfibres from synform hinge (see Fig 2d). (e) Reflected light and (f) cathodoluminescence (CL) images of bedding-parallel slickenfibres (within polished block) and later, cross cutting blocky calcite veinlet. Note that the CL image shows the craters made by the laser during analysis.


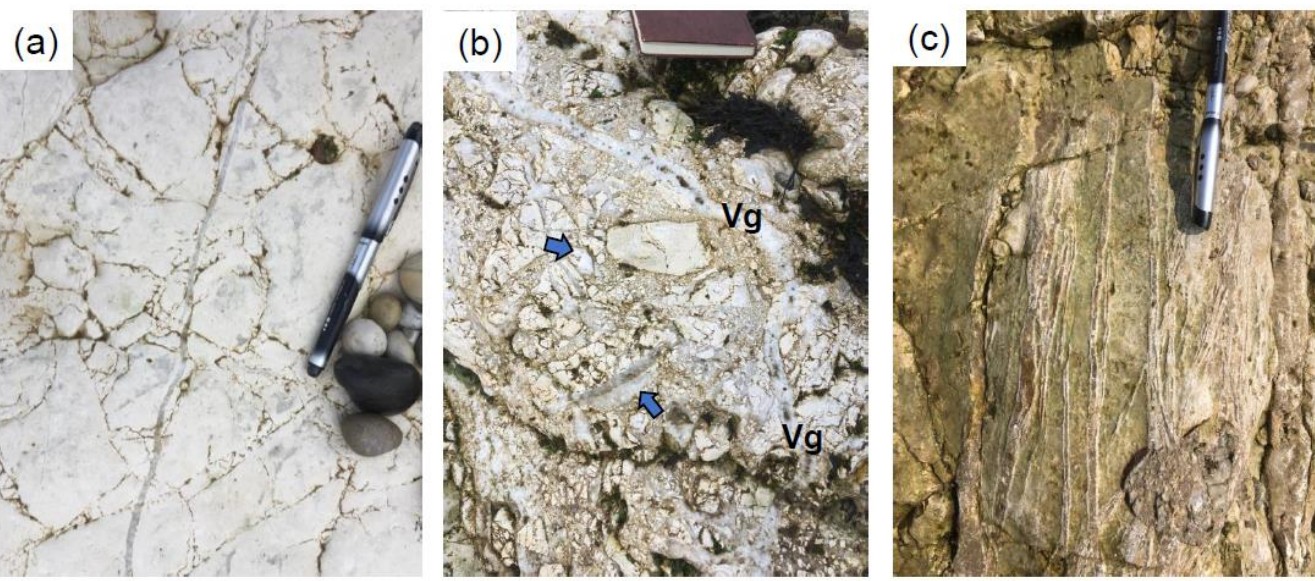

Figure 5. Breccia development associated with extensional movements along the Frontal Faults zone in Selwicks Bay in the
region bounded by the FFS and FFN. (a) Incipient crackle breccia development viewed in plan looking at a bedding plane in
chalk that is cross-cut by a narrow calcite-filled tensile vein. (b) Cross-section view of typical chaotic collapse breccia with
little evidence for shearing or attrition of clasts. Note that calcite veins occur in clasts (examples arrowed) and as cross-cutting
later vuggy features (Vg). (c) Composite calcite veins in plan view from foreshore below high tide with younger orange-stained
ferroan calcite and older white non-ferroan calcite rims.





Figure 6. Brown-coloured marly breccias and sediment fills in tensile fissures associated with the Frontal Fault zone, Selwicks Bay. (a) Marly breccia fill in cross-section view that post-dates sub-parallel calcite veins that line fracture. (b) Marly breccia that pre-dates calcite cemented breccia in plan view. (c) Steeply inclined fissure fill in cross section view that obliquely cross-cuts adjacent calcite veins. (d) Oblique view of irregular subhorizontal zone of fine marly sediment filling the lower part of a fracture that cross-cuts an earlier calcite vein (EV), whilst the upper part of the cavity is filled by a later calcite vein (LV). The sediment is crudely bedded and forms a geopetal fill that youngs upwards, as indicated by the inverted Y symbol.





Figure 7. Thin sections of sediment fills from Selwicks Bay Frontal Fault zone taken in plane polarised light unless indicated otherwise. (a) Chalk wall rocks (WR) and earlier calcite vein fills (V1, V2) unconformably overlain by sediment fill (FF3) cut by slightly later vein (V4); note that the mineral cement in the graded sediment grows in optical continuity with the V4 vein. Note also the line of inclusions separating V1 and V2. (b) Details of calcite-cemented sediment fill showing dark wall-rock clasts, hematite staining, clastic qtz grains (arrowed) and pale sub-angular clasts of earlier calcite. (c-d) As (b), showing included clasts of chert (ch) and glauconite (g) and sponge spicule (arrowed). (e-f) Cockade-style cementation of graded or complex sediment fills where the calcite cements are in optical continuity with the overlying vein fills. (f) is taken with crossed polars.





Figure 8. (a-e) Tera-Wasserburg plots of U-Pb results (all uncertainties shown and quoted at 2σ), and corresponding sample images (see Supplementary Text for full size images). (f) Comparison of the five dates plotted with their 2σ uncertainties.