# Peer review of "Near-surface Palaeocene fluid flow, mineralisation and faulting at Flamborough Head, UK: new field observations and U-Pb calcite dating constraints"

_Solid Earth, 2020_

## Referee Comment (RC1) · Nigel Woodcock (Referee) · 10 May 2020

This paper reports an important scientific contribution to the developing recalibration of the Cenozoic tectonic history of Britain and Ireland. As we currently understand this history, it is driven by two far-field processes, separated in time and only loosely linked kinematically: a) to the northwest, a mantle plume impinging on the developing rift zone destined to become the North Atlantic Ocean; b) to the south, later north-south shortening resulting from the Alpine and/or Pyrenean continental collisions. It is important to assign to one or the other driver the various Cenozoic structures in

[Figure]

Britain. This new work by Roberts et al. demonstrates convincingly that calcite vein formation in the well-studied Flamborough Head Fault Zone dates from the Paleocene, and is therefore linked to the North Atlantic Igneous Province and to the opening of the North Atlantic, rather than to Eocene and later Pyrenean or Alpine events. The science in this paper is sound, and it is presented clearly both in the text and the figures. There are a number of mostly minor suggestions for improvement of the text in the attached annotated pdf. The only substantial suggestion is (at line 52) to discriminate between – as I understand it – the entirely Paleogene Pyrenean collision phase and the mainly Oligocene-Neogene Alpine collision phase.

Please also note the supplement to this comment:
https://www.solid-earth-discuss.net/se-2020-73/se-2020-73-RC1-supplement.pdf

———————————————————

---

## Referee Comment (RC2) · Catherine Mottram (Referee) · 1 Jun 2020

**Catherine Mottram (Referee)**

catherine.mottram@port.ac.uk

Received and published: 1 June 2020

**Summary of paper**

This paper aims to understand the timing of faulting and fluid flow in the Flamborough Head Fault zone in NE England. The authors provide the first absolute timing constraints for carbonate crystallisation using U-Pb geochronology method. Different veins yield dates that span from  $\sim$ 63- 55 Ma. The authors interpret that veins and breccia fills formed largely associated with extensional faulting. It is interpreted that

extensional faulting occurred synchronously with regional igneous dyke intrusions and is associated with the opening of the Atlantic.

General Comments:

This paper is a well-done geochronology study investigating regional timing of faulting and fluid flow. The U-Pb carbonate geochronology method is fast becoming a wellestablished method for directly dating faulting and fluid flow processes in the upper crust. This study uses this novel technique to provide the first absolute timing constraints in the region of interest. This is important because faulting in the Cretaceous chalks is an important process effecting reservoirs across much of Great Britain and the North Sea. This study therefore has the potential to provide a useful insight into the absolute timing and duration of fluid flow and faulting in the region with useful transferable applications to the petroleum and hydrogeology communities. This study is well conceived, uses appropriate methods, produces high-quality geochronology data that is well reported and documented. The interpretations are largely consistent with the data, although the structural history needs to be strengthened. It also would have been nice if the study was broader in scale and scope. Sample documentation could be better represented in the paper. The supplementary material is necessary and supports data in the main paper. Overall I think that this is a useful contribution to understanding the absolute timing of brittle faulting – a topic which is in its' infancy and with some additions (see below and in the annotated version of the manuscript), would make a nice regional geology study suitable for this journal.

Specific Comments:

1. Structural data.

I understand that the authors do not intend to make a detailed analysis of the structural evolution of the area (as stated in lines 158-160), however I think that it would make this paper much stronger if you did include some structural data. That would make the linkage between the dates presented here and the structural interpretation much
stronger and credible. It would also mean that you could interpret the ages relative to the structural setting with more confidence.

I therefore suggest that you add stereonets of the orientation of your samples, the local structures and the regional stress regime (hopefully this shouldn't be too much work as you should already have all the data!)

2. Geological setting

Throughout the writing could be more succinct and you could do a better job of describing the setting without interpreting- make more factual.

A cross section would be useful.

The structure is very linear, could you group together and discuss findings of authors rather than going through study by study.

It would be useful to have a sentence at the end framing your study- why it is interesting and important.

Section 5 might be better coming before section 4, or potentially merged with it. When I initially read section 4, I wanted a lot of the details that are in section 5, so I think restructuring or merging would be beneficial. Some sample numbers on the figures would also be immensely helpful (see my comments below).

3. Link between structural setting, sample description and ages.

Initially it is quite challenging to link together the different structures, photos and samples. I think you do a good job in the supplementary material but in the figures in the text it is less easy to follow. See my comments below on the figures, adding sample numbers, more annotations and a little more context of how the different photographs link together (similar to in the supp material) would be helpful.

4. Abstract - a little more information about the motivations of the study, why it is important and what the significance is would help to attract a broader audience.
5. Throughout make sure that you always keep description/ data reporting and interpretation separate.

**6. Discussion**

Could you provide a (visual?) summary of the relative timing, cross cutting relationships, structural orientation and terminology of vein types? That might help focus your discussion and if you made a figure would be a great visual aid for the reader.

The discussion is OK but you could be a little more definitive about interpreting the timing and sequence of compressional, extensional and strike-slip faults. You could think more about the limitations of your dataset- you have only analysed a few samples, if you broadened out the study it might be possible to fully interpret the timing of the different structures and understand how the regional stress regime has changed through time.

It would be interesting to make some comment about how these different structures might have formed and what overall stress regime you would need in order for the different structures to form.

Addition of some structural data might help you be a little more definitive.

What about pore fluid pressure and interaction between faulting and fluid flow?

How likely is it that there has been multiple periods of extensional faulting – during the Triassic- Cretaceous and then later in the Paleocene as recorded here. Likewise, do you think if you dated more veins that you would end up dating later Cenozoic (re)activation of compressional structures?

See additional comments on annotated PDF of manuscript

Comments on figures:

Figure 1
Please add a key to geological units Consider adding a cross section.

Figure 2:

Please add some structural data (see comments above)- produce stereonets of the orientation of veins related to the major structures.

b) and c) could do with some additional annotation – show where veins are and clearly annotate sample locations

d) it is not clear why you have drawn the arrows on the vug- perhaps a slightly zoomed out image would be more useful for demonstrating that sense of motion.

Figure 3:

a) scale? What evidence is there for the sense of motion drawn in the images? c) Annotations are not clear of folded strata. d) would an additional image taken perpendicular to this one be useful to showing the fold? Throughout the veins could be better annotated and any analysed samples clearly marked.

Figure 4:

How does fig. 4 relate to fig. 3? How do a) and b) relate to each other? Close up of gouge would be useful in b) Clearly label sample labels on analysed samples- are d) e) and f) all the same sample?

Figure 5:

Link back to figure 2 to show where these samples are located Add sample numbers for analysed samples Some closer photographs of textures would be helpful.

Figure 6:

More annotations needed Show where these photographs are on previous photos Add sample numbers for analysed samples

Figure 7:
Add sample numbers More annotations needed F) missing scale.

Figure 8:

The geochronology data and TeraWasserburg plots are good quality- well done!

Ideas for additional figures:

1. Figure clearly showing stereonets with: a) Orientation of main faults b) Orientation of your slickenfibres and veins with respect to these faults. c) You could link to stress analysis done by previous workers (Sanderson and Peacock)?

2. Cross section

3. Better way of linking field photographs (more like you have done in the supplementary material), that is a much clearer way to show how the samples relate to the structures and how each photograph relates to one another.

4. Interpretational diagram synthesising the interpreted vein genesis based on previous work and your ages. This would be a useful visual sum-up of all your data.

Supplementary material

Excellent presentation of methods and data. The only addition could be Concordia plots of secondary standards (Duff Brown and Ash 15) and reproducibility quoted as a %.

Please also note the supplement to this comment: https://www.solid-earth-discuss.net/se-2020-73/se-2020-73-RC2-supplement.pdf Interactive comment

---

## Author Response (AR1)

"This paper reports an important scientific contribution to the developing recalibration of the Cenozoic tectonic history of Britain and Ireland. As we currently understand this history, it is driven by two far-field processes, separated in time and only loosely linked kinematically: a) to the northwest, a mantle plume impinging on the developing rift zone destined to become the North Atlantic Ocean; b) to the south, later north-south shortening resulting from the Alpine and/or Pyrenean continental collisions. It is important to assign to one or the other driver the various Cenozoic structures in Britain. This new work by Roberts et al. demonstrates convincingly that calcite vein formation in the well-studied Flamborough Head Fault Zone dates from the Paleocene, and is therefore linked to the North Atlantic Igneous Province and to the opening of the North Atlantic, rather than to Eocene and later Pyrenean or Alpine events. The science in this paper is sound, and it is presented clearly both in the text and the figures. There are a number of mostly minor suggestions for improvement of the text in the attached annotated pdf. The only substantial suggestion is (at line 52) to discriminate between – as I understand it – the entirely Paleogene Pyrenean collision phase and the mainly Oligocene-Neogene Alpine collision phase."

We thank the reviewer for his positive comments on our study and manuscript.

All minor edits (only a dozen or so) that Nigel suggested on the pdf are adjusted in the revised version, including this comment on Line 52 where we now discriminate between the Pyrenean and Alpine collision.

"General Comments: This paper is a well-done geochronology study investigating regional timing of faulting and fluid flow. The U-Pb carbonate geochronology method is fast becoming a well established method for directly dating faulting and fluid flow processes in the upper crust. This study uses this novel technique to provide the first absolute timing constraints in the region of interest. This is important because faulting in the Cretaceous chalks is an important process effecting reservoirs across much of Great Britain and the North Sea. This study therefore has the potential to provide a useful insight into the absolute timing and duration of fluid flow and faulting in the region with useful transferable applications to the petroleum and hydrogeology communities. This study is well conceived, uses appropriate methods, produces high-quality geochronology data that is well reported and documented."

We thank the reviewer for their positive comments. The rest of the review essentially asks for a more detailed structural study, and for more clarity over the samples and the locations of the samples/photos. As we discuss below, we have not attempted a major structural study, and therefore cannot include this. This was not the aim of this paper. We have made changes to the text and figures/photos so that it is easier to follow how things relate. We note that reviewer 1 did not have any issue with the presentation of our data.

"The interpretations are largely consistent with the data, although the structural history needs to be strengthened." Our aim was never to provide a new structural history that incorporated new structural mapping, as we know from previous work in the area that this would require a level of work akin to a good Masters project at least. As we discuss later on, we are aware that another group is working on revising the structural history and so we deliberately have not gone into further details here.

However, we have made revisions to the text and figures so that 'new' structural observations included here are properly constrained and caveated where necessary. "It also would have been nice if the study was broader in scale and scope."

No comment necessary.

"Sample documentation could be better represented in the paper."

We have improved this. (See later detail).

"The supplementary material is necessary and supports data in the main paper. Overall I think that this is a useful contribution to understanding the absolute timing of brittle faulting – a topic which is in its' infancy and with some additions (see below and in the annotated version of the manuscript), would make a nice regional geology study suitable for this journal."

"Specific Comments: 1. Structural data. I understand that the authors do not intend to make a detailed analysis of the structural evolution of the area (as stated in lines 158- 160), however I think that it would make this paper much stronger if you did include some structural data. That would make the linkage between the dates presented here and the structural interpretation much stronger and credible. It would also mean that you could interpret the ages relative to the structural setting with more confidence. I therefore suggest that you add stereonets of the orientation of your samples, the local structures and the regional stress regime (hopefully this shouldn't be too much work as you should already have all the data!)"

We understand the reviewer's point of view here. Firstly, the comments suggest that the limited structural information we present could be better presented so that readers can find the paper and lines of reasoning easy to follow. Secondly, and most critically, this is a very complicated area with multiple interpretations of the same structures in the literature. Although we made some structural observations, we are aware of another group working in the area, and are aware of them writing up their results (with students involved) at present. It would not therefore be prudent to usurp their work. To be honest, our aim was not to re-characterise the major structures, but to characterise the veining and fluid-flow to an extent that matched the scale of our geochronology, as this aspect of the geology has been much less studied. We know other groups have failed in the past at dating these structures at this particular site, and thus a very large project would be needed to get successful dates from a larger range of structures. Most of our conclusions do not rely on structural observations at the macro or regional scale. We thus do not see the need to add more detail on the regional structures. It is clear from the map that we are located in an E-W trending fault belt, and in our revised version we have avoided speculating too much about motion along this fault belt, or kinematics that created or reactivated it.

Stereonets of samples – A cement in a breccia has no orientation, and a single vug in a vein in a chaotic damage zone has no useful orientation - these are hosted in the E-W trending frontal fault. One other sample is also an E-W trending vein. We feel that adding a stereonet with two E-W trending fault/fracture planes, one vertical and one sub-vertical, is not really going to add much to the paper. The sample from the fold only dates post-folding veining, and thus we do not really discuss the origin of the folding; had we done this, we of course realise that more structural information on the fold orientation would be necessary.

"2. Geological setting: Throughout the writing could be more succinct and you could do a better job of describing the setting without interpreting- make more factual."

Without specific comments is it difficult to ascertain which parts are not succinct enough. We also note that the other review had no issues in this regard. We have not found examples of interpretation in the setting section. We state the timing that previous authors have constrained, these are for example, based on interpretation of seismic; however, that is not us adding interpretation into this section.

"A cross section would be useful."

Spectacularly detailed cross sections – some in 3D - are provided in Starmer 1995 and redrawn in Mortimore (2020); we now point this out. Without reassessing all of the observable structures, we do not feel it is necessary to redraw the same cross-section again.

"The structure is very linear, could you group together and discuss findings of authors rather than going through study by study."

We could, but we choose not to. The other reviewer had no problem here; also, each study generally discussed a different aspect or different methodology, so we feel that the evolution works.

"It would be useful to have a sentence at the end framing your study- why it is interesting and important."

We feel the end of the intro covers this: Here we present data from the Flamborough Head Fault Zone (FHFZ), which forms the southern boundary to the Mesozoic Cleveland Basin, and to which there is no consensus as to the timing and kinematic history. In this paper, we combine new field observations with U-Pb geochronology of calcite veins. Our dates present the first absolute timing constraints on deformation within the FHFZ, and are placed into context with new field observations pertinent to understanding the structural setting of associated fluid flow and fracture filling processes.

"Section 5 might be better coming before section 4, or potentially merged with it. When I initially read section 4, I wanted a lot of the details that are in section 5, so I think restructuring or merging would be beneficial."

Section 4 is very short, and generally describes macro-structures, before section 5 discusses smaller-scale structures. So we feel the order is fine.

"Some sample numbers on the figures would also be immensely helpful (see my comments below)."

Sample numbers are added to additional figure, and other relevant figures.

"3. Link between structural setting, sample description and ages. Initially it is quite challenging to link together the different structures, photos and samples. I think you do a good job in the supplementary material but in the figures in the text it is less easy to follow. See my comments below on the figures, adding sample numbers, more annotations and a little more context of how the different photographs link together (similar to in the supp material) would be helpful."

We have added an additional figure that shows the locations of the samples, as depicted in the supplementary figures. We thought about changing the order of figures, so they occur by locations; however, the text follows a logical narrative, through contractional structures, then extensional structures, then the types of vein-fill. The figures are ordered to match the text, and so we feel they also follow a logical narrative. We have added annotation to the figures to make it clearer what they are intending to show.

"4. Abstract – a little more information about the motivations of the study, why it is important and what the significance is would help to attract a broader audience.

We have added a sentence on why this is important.

"5. Throughout make sure that you always keep description/ data reporting and interpretation separate."

We have been through looking for examples of this. The only examples we can find are where we speculate on the timing of a structure when describing it, this is essentially a field observation. We feel it is OK to speculate something is likely the same age as something else based on its field observations, at the point of describing field observations. It is not really that different to saying structure X cross-cuts structure Y, and therefore X is younger. Yes it is interpretation, but it is generally accepted to make statements like this, and not save such an interpretative statement to later on. Regardless of this, we have moved the field observations so that they come after the dating results, thus they can be read with better context regarding the interpretation.

"6. Discussion Could you provide a (visual?) summary of the relative timing, cross cutting relationships, structural orientation and terminology of vein types? That might help focus your discussion and if you made a figure would be a great visual aid for the reader."

We have chosen not to do this. Each sample comes from a different area of veining, thus, no single figure can capture the cross-cutting relationships. With literally thousands of veins intersecting the damage zone, it would not be an easy task to capture this in a figure and do the outcrop any justice.

"The discussion is OK but you could be a little more definitive about interpreting the timing and sequence of compressional, extensional and strike-slip faults."

We have moved a couple of sentences around, so hopefully this is clearer. Given the limited scope of the structural study carried out here, we also feel that it would be dangerous to be too definitive. Our work sets that scene for further investigations.

"You could think more about the limitations of your dataset- you have only analysed a few samples, if you broadened out the study it might be possible to fully interpret the timing of the different structures and understand how the regional stress regime has changed through time. It would be interesting to make some comment about how these different structures might have formed and what overall stress regime you would need in order for the different structures to form. Addition of some structural data might help you be a little more definitive. What about pore fluid pressure and interaction between faulting and fluid flow?"

We feel these are rather general review comments. We feel our discussion already discussed the limitations of the dataset, e.g. ". It should be noted that we cannot rule out that fluid-flow and tensile fracturing may have extended to even younger dates than our study implies."

We are not sure what the reviewer means by broadening out the study. Study a larger area, use different techniques?... These comments all seem to allude to a structural re-interpretation, which as we have already mentioned, was not our motivation. We do however point out that our results suggest this would be fruitful in terms of improving our regional understanding. Regarding pore fluid pressure, here too we feel that given the limits of what is presented and the relatively narrow scope of our sampling, it would require undue amounts of speculation on our part to go into much detail about this. The preservation of sediment fills and open vugs is somewhat problematic for the generation of overpressures as this generally requires a sealing mechanism to occur. In essence, we feel that this issue falls outside of the scope of the present study.

"How likely is it that there has been multiple periods of extensional faulting – during the Triassic- Cretaceous and then later in the Paleocene as recorded here."

The Chalk host rock here is Cretaceous, and so we are not sure of the relevance of this question. The fault zones described here may overlie and reactivate an older region of basement faulting at depth, but since we do not attempt to provide new information on why the FHFZ formed, we have not discussed this in any detail. Nor do we feel – in the context of the present paper and its findings – that this is necessary.

"Likewise, do you think if you dated more veins that you would end up dating later Cenozoic (re)activation of compressional structures?"

No. We think the extensional faulting is younger than the folding, and we see no evidence for later reactivation of these structures. There is also little material to actually date the compressional structures, these require syn-folding or syn-thrusting slickenfibres, and these are not that common. Those that do occur appear to not contain enough U to allow dating. This is one of the joys of calcite dating!

"See additional comments on annotated PDF of manuscript"

"Comments on figures: Figure 1 Please add a key to geological units Consider adding a cross section"

See earlier comment on cross section. Key added.

"Figure 2: Please add some structural data (see comments above)- produce stereonets of the orientation of veins related to the major structures. b) and c) could do with some additional annotation – show where veins are and clearly annotate sample locations d) it is not clear why you have drawn the arrows on the vug- perhaps a slightly zoomed out image would be more useful for demonstrating that sense of motion."

See previous comment on steronets and structural data. The relationship between veining and major structures is described perfectly clearly in the text. We have added additional annotation to the figures. The arrows show the extension direction (Mode 1), we have labelled this.

"Figure 3: a) scale? What evidence is there for the sense of motion drawn in the images? c) Annotations are not clear of folded strata. d) would an additional image taken perpendicular to this one be useful to showing the fold? Throughout the veins could be better annotated and any analysed samples clearly marked."

Offset beds show the sense of motion. We have added further annotations. There are no samples from this locality.

"Figure 4: How does fig. 4 relate to fig. 3? How do a) and b) relate to each other? Close up of gouge would be useful in b) Clearly label sample labels on analysed samplesare d) e) and f) all the same sample?"

Sample is labelled on figure 2. The first picture is replaced with another that is a better close-up of the gouge material.

"Figure 5: Link back to figure 2 to show where these samples are located Add sample numbers for analysed samples Some closer photographs of textures would be helpful."

These are all from the foreshore in front of the frontal faults, this is now stated. No samples came from these outcrops.

"Figure 6: More annotations needed Show where these photographs are on previous photos Add sample numbers for analysed samples."

No samples came from these outcrops. Photos are from the damage zone of the Frontal Faults, this is now stated. Annotation added.

"Figure 7: Add sample numbers More annotations needed F) missing scale."

Photos are not related to samples, scale fixed.

"Figure 8: The geochronology data and TeraWasserburg plots are good quality- well done!"

"Ideas for additional figures: 1. Figure clearly showing stereonets with: a) Orientation of main faults b) Orientation of your slickenfibres and veins with respect to these faults. c) You could link to stress analysis done by previous workers (Sanderson and Peacock)? 2. Cross section 3. Better way of linking field photographs (more like you have done in the supplementary material), that is a much clearer way to show how the samples relate to the structures and how each photograph relates to one another. 4. Interpretational diagram synthesising the interpreted vein genesis based on previous work and your ages. This would be a useful visual sum-up of all your data. Supplementary material Excellent presentation of methods and data. The only addition could be Concordia plots of secondary standards (Duff Brown and Ash 15) and reproducibility quoted as a %. Please also note the supplement to this comment: https://www.solidearth-discuss.net/se-2020-73/se-2020-73-RC2-supplement.pdf"

Our previous responses cover these comments.

We have been through the attachment and made minor alterations to the text where we saw room for improvement.

[revised manuscript text omitted]

(a)

(b)

(c)

(d)

(e) Reflected light                                    Host rock (f)                              Slicken fibres    Cross-cutting veinlet

CL

[Figure]

Figure 5. (a-e) Tera-Wasserburg plots of U-Pb results (all uncertainties shown and quoted at 2σ), and corresponding sample images (see Supplementary Text for full size images). (f) Comparison of the five dates plotted with their 2σ uncertainties.

[Figure]

Figure 64. Structures associated with contractional structures in Selwicks Bay; all from locality 3. (a) Crush breccia along top-to-the-S thrust fault reoriented by later, normal fault-related tilting. (b) Crush breccia and brown gouge derived from chalk and shale, respectively, associated with top-to-the-S thrust fault with narrow gouge injections into footwall, one of which is arrowed.

(c) Calcite-hematite slickenlines associated with top-to-the-S thrust fault.

[Figure]

Figure 75. Breccia development associated with extensional movements along the Frontal Faults zone in Selwicks Bay in the region bounded by the FFS and FFN; all photos are from the foreshore.. (a) Incipient crackle breccia development (plan view)viewed in plan looking at a bedding plane in chalk that is cross-cut by a narrow calcite-filled tensile vein. (b) Cross-section view of typical chaotic collapse breccia with little evidence for shearing or attrition of clasts. Note that calcite veins occur in clasts (examples arrowed) and as cross-cutting later vuggy features (labelled vugVg). (c) Composite calcite veins in plan view from foreshore below high tide with younger orange-stained ferroan calcite and older white non-ferroan calcite rims.

[Figure]

[Figure]

Figure 86. Brown-coloured marly breccias and sediment fills in tensile fissures associated with the Frontal Fault zone, Selwicks Bay; photos a and c from the cliff, and b and d from the adjacent foreshore. (a) Marly breccia fill in cross-section view that post-dates sub-parallel calcite veins that line fracture. (b) Marly breccia that pre-dates calcite cemented breccia in plan view. (c) Steeply inclined fissure fill in cross section view that obliquely cross-cuts adjacent calcite veins. (d) Oblique view of irregular subhorizontal zone of fine marly sediment filling the lower part of a fracture that cross-cuts an earlier calcite vein (EV), whilst the upper part of the cavity is filled by a later calcite vein (LV). The sediment is crudely bedded and forms a geopetal fill that youngs upwards, as indicated by the inverted Y symbol.

[Figure]

[Figure]

Figure 97. Thin sections of sediment fills from Selwicks Bay Frontal Fault zone taken in plane polarised light unless indicated otherwise. (a) Chalk wall rocks (WR) and earlier calcite vein fills (V1, V2) unconformably overlain by sediment fill (FF3) cut by slightly later vein (V4); note that the mineral cement in the graded sediment grows in optical continuity with the V4 vein. Note also the line of inclusions separating V1 and V2. (b) Details of calcite-cemented sediment fill showing dark wall-rock clasts, hematite staining, clastic qtz grains (arrowed) and pale sub-angular clasts of earlier calcite. (c-d) As (b), showing included clasts of chert (ch) and glauconite (g) and sponge spicule (arrowed). (e-f) Cockade-style cementation of graded or complex sediment fills where the calcite cements are in optical continuity with the overlying vein fills. (f) is taken with crossed polars.

[Figure]

Figure 8. (a-e) Tera-Wasserburg plots of U-Pb results (all uncertainties shown and quoted at 2σ), and corresponding sample images (see Supplementary Text for full size images). (f) Comparison of the five dates plotted with their 2σ uncertainties.